# Assessment of Natural Transmission of Bovine Leukemia Virus in Dairies from Southern Chile

**DOI:** 10.3390/ani12131734

**Published:** 2022-07-05

**Authors:** Bibiana Benavides, Gustavo Monti

**Affiliations:** 1Animal Health Department, University of Nariño, San Juan de Pasto 52001, Colombia; bbenavides@udenar.edu.co; 2Quantitative Veterinary Epidemiology Group, Animal Sciences Department, Wageningen University and Research, 6702 PB Wageningen, The Netherlands

**Keywords:** bovine leukemia virus, cattle, management, risk factors

## Abstract

**Simple Summary:**

A longitudinal study was conducted to describe the frequency and epidemiological association of risk management practices related to new cases of BLV in cattle on dairy farms in Southern Chile. Animal information was obtained from the records of each farm, as well as blood and milk samples that, tested by commercial blocking ELISA to assess the infection status of animals. A higher number of new cases was found in adult animals that were related to practices, such as rectal palpation, artificial insemination, and injections. However, it is important to carry out other studies that establish the role of management practices in the spreading of BLV to improve the control of disease.

**Abstract:**

Bovine leukemia virus (BLV) is a retrovirus that affects cattle worldwide. A longitudinal study was conducted with the aim to (a) estimate the incidence rate of the BLV infection of dairy farms in the regions of Los Ríos and Los Lagos (Chile), and (b) describe the frequency and epidemiological association of risk management practices related to new cases in cattle on dairy farms in Southern Chile. Infection status was based on commercial blocking ELISA results, on serum and milk. Individual information on animals and management practices was extracted from farm records, and then the most likely date of infection for new cases was estimated. The number of new infections was used to calculate the within-herd incidence rate. Adult animals had an incidence rate of 1.16 (95% CI 0.96; 1.20) cases per 100 cow-months at risk, while for young animals it was 0.64 (95% CI 0.44; 1.00) cases per 100 animal-months at risk. Rectal palpation, artificial insemination, and injections were the most common practices related to infection. Further studies are needed to determine if these are the only practices that facilitate spreading or if there are other practices that can be handled better in order to reduce the spread of BLV.

## 1. Introduction

Bovine leukemia virus (BLV) is a retrovirus that affects cattle, producing a lymphoproliferative disorder that includes lymphocytosis, malignant lymphoma, and lymphosarcoma [1]. BLV can cause enzootic bovine leukosis (EBL); the disease has a worldwide distribution and generates a high impact on the cattle industry, especially in dairy cattle, in which there is a greater incidence due to management practices and the time of the permanence of the animals on the farm [2]. BLV impacts negatively on animal welfare and could generate direct economic losses by impairing host response to vaccination [3], reducing cow longevity [4,5], decreasing milk production [4,6], and restricting trade of embryos and live cattle [7]. 

BLV transmission occurs via exposure to infected lymphocytes [8] and can be transmitted from mother to calf during pregnancy or through colostrum of infected dams [9,10]. Horizontal transmission is the most common via iatrogenic routes in group management practices, such as blood-contaminated needles, surgical instruments, or gloves for rectal examination [11,12,13]. Control strategies include guiding farmers about changes and a reduction or elimination in practices that facilitate transmission of BLV.

European countries, including the UK, France, Germany, Spain, Sweden, Norway, Denmark, and the Netherlands, are officially free from EBL [7]. Other countries present a contrasting epidemiological situation with a large prevalence of EBL, such as the USA [14,15,16], Argentina [17], Canada [18], and Japan [19,20]. 

In Chile, the situation seems to be more favorable. The most recent study estimated an overall proportion of infected herds of 34.7% [21] and an overall seroprevalence of 5.3%. However, there are significant differences in the prevalence of infected herds and the within-herd prevalence between herds of different sizes [21] due to substantial differences in management practices. Farms vary from large commercial herds to small ones with few cows for subsistence. 

Milk production in Chile is an important economic activity, and for many small dairy farmers, a significant livelihood. Although milk producers are distributed throughout the country, the largest supply to the dairy industry (>80%) is produced in the southern regions (Los Ríos and Los Lagos) [22], where 84.2% of the total number of dairy cows is concentrated [23].

Most of the studies to identify the risk factors associated with BLV are cross-sectional studies and estimate the prevalence of the disease [2]. However, prevalence studies reflect the proportion of cows infected during a period and assume that transmission occurred at an unspecified time in the past. A better indicator is the incidence, which is the estimated number of new infected cows throughout the study period and allows for inclusion in the estimation, the time period in which the transmission occurred; furthermore, this indicator is not affected by the duration of the infection [24]. Therefore, estimating the incidence associated to risk management practices will be a valuable tool for disease control.

Thus, the objectives of this study were: (i) to estimate the incidence rate of BLV in dairy farms in the Los Ríos and Los Lagos regions, and (ii) to describe the frequency and epidemiological association of risk management practices that could be related with new cases.

## 2. Materials and Methods

### 2.1. Location and Population

This study was conducted in the southern zone of Chile (Los Ríos and Los Lagos regions). Eleven farms were enrolled for convenience, based on background information about infected farms that were sampled in a previous study of BLV in Southern Chile [21], the willingness of the owners to collaborate in the study and farms with permanent and updated records of group management practices. This sample consisted of 3 small (<40 animals), 2 medium (between 41 and 200 animals) and 6 large (>200 animals) farms, this classification according to the criteria adopted by the National Institute of Statistics (INE). For the estimation of the incidence rates all animals older than six months were used. For the assessment of the association of management practices with new infection, the population under study was cows in lactation, given the diagnostic and management possibilities.

In total, 9544 serum samples were obtained from 2450 bovines (older than six months), but 146 were not used because the animals were present only in one sample, and no incidence could be estimated. The 2304 remaining animals corresponded to adult cows and some bulls (66.2% (*n* = 1527)); 33.8% (*n* = 777) were female youngstock. In addition, we obtained 7445 milk samples from 1527 lactating cows. All animals had individual identification systems.

### 2.2. Experimental Design

We conducted a longitudinal prospective study over 18 months to estimate the incidence rate of BLV in cattle on dairy farms in Southern Chile.

An infected animal was defined as an individual that was positive either to the ELISA test in serum or milk for at least two consecutive samplings, and a new infection was considered for an animal that tested positively for the first time to the ELISA test after being negative in the previous testing.

### 2.3. Follow-Up Scheme

Individual information on the animals was extracted from farm records, including date of birth, management practices that could pose a risk of transmission, such as palpation, insemination, or direct mounting, vaccinations, injections, and blood sampling. 

Given the dynamic of the dairy populations, during each visit, we identified new animals to be included in the study. Therefore, the follow-up scheme for determining the health status of the dairy cows is depicted in Figure 1; young stock animals and bulls were serologically tested at the beginning and end of the follow-up period. To avoid needless repeated bleeding and because both tests have similar good sensitivities, we employed serum and milk ELISA assays in adult cows [25]. 

Blood samples were taken by venipuncture from the medium coccygeal vein, yielding 6 mL, and kept refrigerated until they arrived at the Universidad Austral de Chile’s laboratory. The samples were centrifuged upon arrival, and the serum was kept at −20 °C in Eppendorf tubes until processing. Milk samples were taken during milking, with 5 mL taken during the initial milk squirts and maintained in a refrigerator until arrival at the laboratory, where they were then held at −20 °C until ELISA processing.

### 2.4. Serum and Milk ELISA Tests

We tested serum and milk samples using commercial blocking ELISA kits (INGEZIM BLV COMPAC 2.0) based on the gp51 monoclonal antibodies, following the protocols recommended by the manufacturers. The test allows the detection of specific antibodies either in bovine serum or milk. 

### 2.5. Data Analysis and Processing

We estimated the incidence rate (IR), which is defined as the number of new infections in a population per unit of animal-time during a given time period (a month for this study) [24]. The denominator was calculated by the exact method; therefore, we built an array with the exact amount of animal-time at risk contributed by each animal in the study population, which also considered the inputs (births and buying of animals) and outputs (dead and cullings).

In addition, we compiled data using information from each farm’s records, and then estimated the most likely date of infection for new cases. For the age of presentation for new infection, we used a descriptive statistical analysis that included Shapiro–Wilk testing to evaluate data normality and then a non-parametric Kruskal–Wallis association test with 95 percent confidence. 

The number of new infections was used to calculate the incidence rate, which was then modelled as a within-herd incidence rate. For animals who tested negative in the first sample, time-at-risk was computed starting with initial sampling (left censoring) and ending at the estimated time of infection, as detailed below, or right-censored with the study’s end. Stata^®^ (version 9.0) was used to perform the calculations. Because ELISA tests could not discriminate a positive reactor due to colostral antibodies from active infection in calves younger than six months, they were removed from the study.

Because the time at which each new infection occurred was unknown, we estimate the possible time of infection following [26]. Briefly, it starts from the days that elapsed between the seven days before the last observation with a negative serological result (*t*_1_) and the first observation with a positive serological result (*t*_2_), using a Bayesian approach based on the concept of census intervals for survival analysis [27]. Let $T_{\alpha }\;$ represent the time till seroconversion for an animal after the start of follow−up and assume that $T_{a}\;\in \;\left({T^{\prime }}_{1},\;{T^{\prime }}_{2}\right)$. Then, if θ  represents the current time till infection, it follows that
(1)$$Pr\;\left(T_{\alpha }\in \left({T^{\prime }}_{1},{T^{\prime }}_{2}\right)|\theta \right)=Pr\left(T_{\alpha }-\theta \in \left({T^{\prime }}_{1}-\theta ,{T^{\prime }}_{2}-\theta \right)|\theta \right)$$

The quantity $T_{\alpha }-\theta$ is the time till seroconversion from experimental infections, which was derived from a previous study [8]. Assuming that $T_{\alpha }-\theta$ has a distribution with survival function St, then (1) is:(2)$$Pr\left(T_{\alpha }\in \left({T^{\prime }}_{1},{T^{\prime }}_{2}\right)|\theta \right)=S\left({T^{\prime }}_{1}-\theta \right)-S\left({T^{\prime }}_{2}-\theta \right)$$
and follows a gamma distribution. We reflect prior knowledge or uncertainty about θ in the form of a density *D*(θ), where D(θ) ≥ 0  and $\int_{0}^{\infty }D\left(\theta \right)d\theta =1$. Then the posterior density for θ, L<θ≤ U, where L and U are the lower and upper bounds of the time till infection given the data, takes the following expression:(3)$$D\left(\theta |Tser\;\in \;\left(t_{1},t_{2}\right)\right)=D\left(\theta \right)\left\{S\left(t_{1}-\theta \right)-S\left(t_{2}-\theta \right)/\int_{L}^{U}D\left(\theta \right)\left\{S\left(\left(t_{1}-\theta \right)\right)-S\left(\left(t_{2}-\theta \right)\right)\right\}\right\}d\theta$$

We obtain the point estimate of θ by finding the value $\theta^{\prime }$ or the median value of many, which maximizes the function, and we consider it the estimate of the most probable time till infection. From this period estimated in days, we obtained a lower and upper limit and applied to the date of seroconversion to obtain a probable date of infection and establish the time of occurrence of new infections and the description of management practices carried out during that period. Calculation was obtained using Winbugs V. 1.4.3 [28].

Associations between each risk management practice and new infection were tested using a Chi square test (*p* < 0.05).

## 3. Results

### 3.1. Overall Results

During the study, we obtained 2450 serum samples and another 1527 milk samples from cows in production. Based on the ELISA test results, 963 cows tested negative and 459 cows (32.3%; 95% CI (29.8; 34.8)) were infected; 105 (22.9%) were new infections. For the group of young animals (*n* = 777), 697 tested negative, 58 (10.3%; 95% CI (8.2; 12.6)) tested positive, and 22 (27.5%) were new infections.

### 3.2. Incidence Rate

Adult animals had an incidence rate of 1.16 (95% CI 0.96; 1.20) cases per 100 cow-months at risk, while young animals had an incidence rate of 0.64 (95% CI 0.44; 1.00) cases per 100 animal-months at risk. 

The incidence rate for young and adult animals by farm is summarized in Table 1.

Farm F had the largest rate of adult animal incidence, with 2.92 cases per 100 cow-month at risk, and 60% of the animals were infected. The farm with more new cases (*n* = 40) and an incidence rate of 2.12 cases per 100 cow-month at risk was Farm I. During the follow-up period, Farm K had no new cases. 

Only six farms had new cases of young animals to report (heifers younger than 18 months and calves over 6 months). Farm F had the greatest incidence rate (2.04 cases per 100 cow-month at risk), as well as the highest number of cases (*n* = 11). 

The incidence rate was not calculated on farms B, C, H, and K because the young animals lacked a suitable identifying system to track them after birth. Furthermore, during the follow-up period, farm J did not exhibit any new cases.

### 3.3. Description of New Cases and Management Practices

The age of presentation of the new infections did not present a normal distribution (Shapiro–Wilk test *p* < 0.05) and was composed of individuals of all ages, with a higher concentration between 2 and 6 years (Figure 2).

The median age for new infections was 4.2 years, and the interquartile range was from 3.2 to 6. The group with the highest number of new infections (*n* = 25) was that of 3–4 years old, followed by animals older than 6 years (*n* = 22) and 1–3 years (*n* = 20). Only four animals were younger than two years or first birth; the difference between the groups is statistically significant (*p* < 0.0001).

The most frequent management practices recorded were rectal palpation, insemination, direct mounting (used only on three farms), injections, and vaccinations. 

From the 105 new infections identified in the study, we obtained information on age and management practices during the study period for 94 cases (Table 2). Based on the estimation of the most likely time of infection, in 33 new cases, we could relate any of those risk practices.

Rectal palpation, artificial insemination, and injections were the most common practices related to infection (*p* < 0.05). Direct mounting was practiced on just three farms, and on the farm where new cases were identified, the differences were not statistically significant (2 = 73.5, *p* > 0.05) to be related to this procedure, but they did occur within the likely time of infection.

## 4. Discussion

The present study was carried out on the population of cows in production for the feasibility of monitoring through ELISA in milk to describe the natural transmission of BLV in Southern Chile through a measure of disease occurrence.

The overall incidence rate for adult cattle was 1.16 cases per 100 cow-months at risk, which is lower than rates recently reported in the United States [2] and Argentina [29]. However, as those studies showed, there is a large variation in the incidence rates amongst herds. For example, the incidence rate in adult animals was higher for the large farms (F, G, and I), which matches to a prior study in which the prevalence of these farms was 34.8 percent and the prevalence of small farms, such as farm D, was 2.1 percent [21]. These higher rates in herds with higher prevalence have been observed in other studies [2,29], showing that high prevalence is a risk factor for effective transmission of the virus [2,30], and infection force is density dependent [31]. The drop in the rate following the removal of infected animals [26,32,33], which is employed as a method to control the spread in infected herds, is another indicator for the influence of the proportion of infection on the incidence rate. Again, we saw a trend in that direction in the study. However, the follow-up period was brief and the number of herds was small, making it difficult to examine this association more clearly and statistically. 

From another study [29], the incidence rate in young animals was lower (0.64 cases per 100 cow-months at risk) than in adult animals, even though the interval between infection and seroconversion is shorter in young animals [32]. However, as pointed out, the importance of including young animals when designing and implementing control programs should not be understated [34,35].

Most new cases occurred in bovines aged one to four years and those aged six years under the study’s conditions. This variation in the age of infected animals could be due to different routes of transmission or different types of management practices that allow for more significant contact between animals of different ages when animals enter the reproductive age and are exposed to older cattle, many of them with a high proviral load [26,36]. In addition, around parturition, the antibody transfer from peripheral blood to colostrum together with physiological hormonal changes in this period may be beneficial to the reactivation of viral expression in BLV-infected cows, thus, increasing their potential infectivity, postulated as more efficient transmitters [37]. It does not necessarily indicate a different susceptibility due to age or a relationship between age and viral load, as other studies have attempted to establish and failed to do [38]. One limitation of this study was that we did not include young calves that could also be infected and spread the virus given that vertical transmission is documented [39,40].

Epidemiologic studies on dairy cattle have demonstrated an association between BLV infection and decreased productive lifespan within herds [4,41,42], although others have not found this association [5,43]. Therefore, those findings should be further investigated in future studies for different cattle breeds, geographic areas, and management systems for animals with high vs. low proviral load.

Several routine management practices have been implicated in BLV transmission [44,45,46]. In this study, rectal palpation was the management practice most associated with the occurrence of new cases. Transmission of BLV via rectal exam sleeves has been demonstrated in both experimental [47] and observational studies [12,45,48], and it is an extensively used practice in dairy but also in beef farms in Chile and many countries [36].

Another practice worth mentioning is direct mounting, which occurred within the estimated time of infection for three individuals from the same farm, where the bulls were found to be BLV reactors. Transmission is linked with virus-infected lymphocytes in the genital tract of the bull and then transferred during copulation in these circumstances [39,49], although a more recent study suggests a potential role of the smegma from infected bulls [50]. In addition, other studies did not find a higher risk when bulls are aleukemic or have a low proviral load [51,52]. However, BLV has been detected in semen straws or by apparent contamination with blood from connected glands [50,53,54]; nevertheless, transmission by this route has yet to be demonstrated and it is thought as rare. As a result, artificial insemination centers should ensure that donor bulls are BLV free and reactor negative [55].

In all participating farms, injection and vaccination protocols are documented regularly and extensively and six cases were linked to that occurrence in the current analysis. However, we were unable to track the order in which the animals were injected, which is regarded as sciential aspect in transmission [56]. Furthermore, there is little evidence in the literature that the transmission of these practices is effective. Some studies have found that the amount of blood delivered in a needle is enough to cause infection [39,57], but when the recipient animal is persistent lymphocytic or with high proviral load [33,58]. On the other hand, other researchers find that the amount of blood required for successful transmissions, such as through an intradermal needle, is insufficient [59], even from bovines with high proviral doses. Furthermore, results from previous studies were obtained using experimental designs that were not the same as the current ones, which were based on commercial records [56,60].

Even though most massive events are recorded in the selected farms, establishing a link between these management practices and the incidence of new cases proved difficult. Other techniques, such as parenteral treatments during milking, are carried out in small groups and are not recorded. To establish a valid correlation for these variables, it is also necessary to identify how the animals enter each activity and to verify that an infected animal went before a susceptible animal. Thorough investigation of specific high-risk herds would enable the elucidation of important risk factors for within-herd transmission and the origin of the new cases considering disease transmission varies among farms, and study design varies in the ability to identify the true significance and direction of associations.

Some of the study’s limitations include the possibility that some cows were new cases during the peripartum period, whether cows had physiological immune suppression [61], and the fact that we used a serological test to assess infection status, that resulted in a negative test result, which was then linked to other management practices. Another limitation of our study is that the follow-up period was too short to discover seasonal effects in transmission rates, such as reduced transmission rates in the winter [32] or higher transmission rates in the summer [62,63], which could be related to the activity of blood-feeding insects. Finally, another limitation is the different precision in the incidence rates between age groups because the young stock was only sampled twice with blood samples, and the adult cows had multiple milk samples and two blood samples.

Assessing the potential association between management practices and natural transmission in commercial farms is challenging. Even though the management procedures of the selected herds were well documented, it cannot be ruled out that BLV infection was produced through iatrogenic transmission or transmission by other causes; all this resulted in several new infections that were not possible to relate to any practice. However, some bias could also occur given that we used records from commercial farms, and there is variability in recording the management practices for animal records. 

## 5. Conclusions

This study estimated the incidence rate for adult animal (1.16 (95% CI 0.96; 1.20) cases per 100 cow-months at risk) and young animal (0.64 (95% CI 0.44; 1.00)) cases per 100 cow-months at risk and identified management practices associated with BLV seroconversion incidence rate among cows on Chilean dairy farms. The most frequent management practices were rectal palpation, artificial insemination, direct mounting, injections, and vaccinations. However, Further studies are needed to determine if these are the only practices that facilitate spreading or if there are others, which can be handled better in order to reduce the spread of BLV.

## Figures and Tables

**Figure 1 animals-12-01734-f001:**
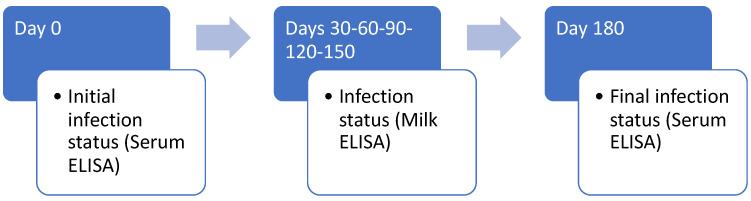
Follow-up process for identification of new infected cases in adult cows.

**Figure 2 animals-12-01734-f002:**
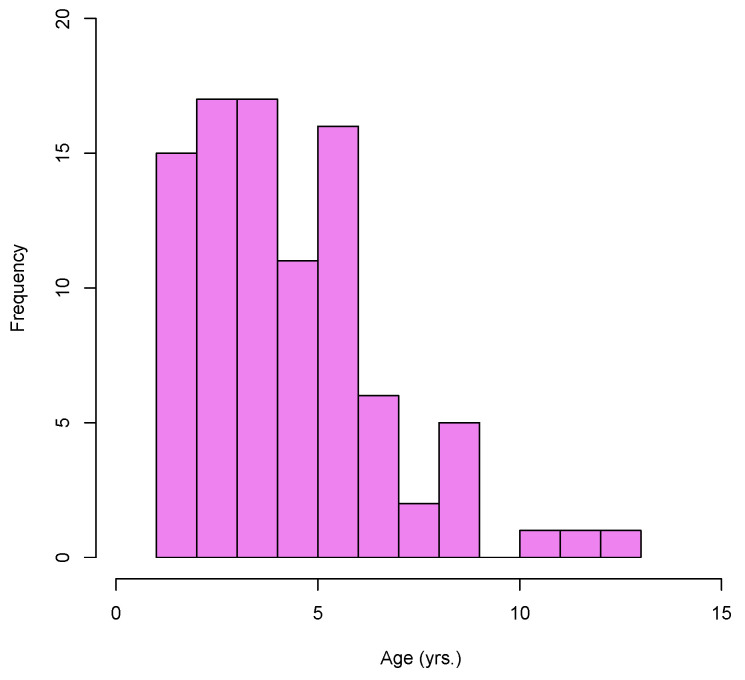
Histogram representing age distribution of new cases detected in cattle from dairy farms in Southern Chile.

**Table 1 animals-12-01734-t001:** Incidence rate (expressed in infections per 100 animal-month at risk) and its 95% confidence interval of BLV infections, for the groups Adult Cows and Young Animals, by Farm.

Farm	Adult Cows						Young Stock					
	NI (n)	IR	(95% CI)	I (n)	Sus	Total	NI(n)	IR	(95% CI)	I (n)	Sus	Total
A	4	1.28	(0.48; 3.48)	1	24	29	1	1.48	(0.20–10.56)	0	12	13
B	1	1.76	(0.01; 1.24)	8	54	63	0	ND	ND	1	53	54
C	1	0.24	(0.01; 1.92)	5	23	29	0	ND	ND	0	11	11
D	1	0.36	(0.04; 2.72	10	19	30	1	0.88	(0.12; 6.48)	0	30	31
E	13	0.76	(0.44; 1.28)	92	114	219	3	0.34	(0.08; 1.04)	0	126	129
F	16	2.92	(1.80; 4.80)	89	46	151	11	2.04	(1.12; 3.72)	0	87	98
G	26	1.32	(0.88; 1.96)	92	252	370	3	0.52	(0.16; 3.12)	41	67	111
H	1	0.04	(0.00; 0.48)	7	225	233	0	ND	ND	0	72	72
I	40	2.12	(0.39; 2.92)	139	113	292	3	1.00	(0.81; 3.12)	0	161	164
J	2	1.16	(0.28; 4.68)	15	65	82	0	0.00	ND	16	73	89
K	0	0.00	ND	1	28	29	0	ND	ND	0	5	5
**Total**	**105**			459	963	**1527**	**22**			58	697	**777**

NI = New infections; I = Infected; NoInf = Susceptible; ND = Not calculated.

**Table 2 animals-12-01734-t002:** Description of the number of new infections of Bovine Leukemia Virus and the number of those that were associated to some potential BLV transmission risk management practices.

Age	New Infections		New Cases by Risk Management Practices Related to Transmission (n)			
(Years)	n	%	Mounted	AI	Palpation	Injections
1–3	20	21.3	0	2	0	2
3–4	25	26.6	1	3	5	0
4–5	17	18.1	0	3	5	2
5–6	10	10.6	0	1	1	1
>6	22	23.4	2	2	2	1
Total	94	100.0	3	11	13	6

AI = Artificial Insemination.

## Data Availability

The data presented in this study is available on request from the corresponding author.

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
