# Peer review of "Assessment of Natural Transmission of Bovine Leukemia Virus in Dairies from Southern Chile"

_animals, 2022, doi:10.3390/ani12131734_

Round 1

Reviewer 1 Report

Bovine leukosis virus (BLV) is an important retrovirus that affects cattle worldwide. In this study, the authors estimated the incidence rate of the BLV infection in dairy farms in southern Chile and analyzed the frequency and epidemiological association of risk management practices related to the clinical cases. The results is interesting, and will be useful for the prevention of BLV. But the manuscript was not well organized, drafted and several issues should be revised.  

1. line 50, whats the EBL means?

2. The first paragraph of section 2 and section 3 should be titled.  

3. line 93-94, whats animals mean, and whats the difference between animals and cows?

4. line 93, what s the differences for whole blood and serum?

5. Section 2.1 and Figure1, why did two types of samples, serum and milk, were selected? If the author want to avoid needless repeated bleeding, milk samples could be selected for the whole sampling process.

6. Line 123-151, the description of the operating principle, protocol and criteria for results judgement seems not necessary and could be deleted or shortened into one-two sentences.

7. Line 199-203, how many adult cows were sampled? How many young animals were sampled?

8. Line 205, whats the cow-months” means?

9. The use of case was not appropriate, a case usually means the occurrence of a disease in a herd or farm, but according to the authors definition, the case means one newly positive animal. So I suggest the use of negative, positive, infected to describe different infection conditions.

10. The aim of the study is to assessment the transmission of BLV in the different farms, the author use ELISA antibody results to identify the infection status, especially the newly infected ones. So the animals should be negative at the beginning, the positive animal is useless for this study. The author should focus on the negative- positive individuals.

11. Table 1, the author should include the number of negative and positive animals in Table.

12. There are grammatical errors for some sentences, and the expression of lots of sentences were not clear and unintelligible. Such as: Line 41-44, but also is not corrected used; line 65-67; line 86-89; line 93-94; line 213, should be “60% of the animals were infected”; line 215, should be “Farm K had no new case.; etc.

13. line 231, of this group means which group?

14. line 232, it could not be concluded four animals were younger than two years or first birth from Table 2.

15. Table 2, statistical analysis results should be shown.

16. line 243, how did the author conclude “Rectal palpation, artificial insemination, and injections were the most common practices related to infection”?

17. line 107-109, which samples were collected for young animals and adult animals?

18.line 237-239, only 33 cases (~1/3 of the total) were related to one of the selected risk practices. How about the possible reasons for the other 2/3 cases?

Reviewer 2 Report

An interesting article that can add some information to the current knowledge regarding BLV transmission in countries around the world.  However, the manuscript does not contain sufficient information for a reader to fully understand how the study was conducted and some useful information is missing from the results.

Additionally, I strongly recommend employing an English proofreading and copyediting service; I have not listed grammar errors and typos unless absolutely necessary, as they are numerous.

Abstract

The simple summary and scientific abstract are almost identical, and it is doubtful that a lay person would completely understand the simple summary.  At the editor’s discretion, I recommend rewriting the simple summary to make it more accessible to someone inexperienced in BLV research.

Line 19-20: sentence starting with “Further studies are required…” is very awkwardly worded; suggest revising.  Same comment for lines 32-33.

Line 21: BLV is most commonly referred to as “bovine leukemia virus,” not “bovine leucosis virus;" suggest changing this to more accurately reflect common terminology used in most literature.  Same comment for line 37.

Introduction

Currently, this is not well organized and it is difficult for the reader to follow logically from the introductory material to the main objectives of the study.  There is a lot of general introductory material, but how this relates to the specific situation in Chile is less clear, and it’s hard to link all of the introduction to the specific aims of this study.

Line 37:-38: please be more specific in indicating which species BLV infects, rather than just stating it infects “individuals”

Line 50: you have used the acronym EBL but have not defined it yet.  Please either explain what EBL is or change it to BLV.

Line 50-52: this sentence is confusing – what is being estimated in these studies cited?

Line 55-56: what are these significant differences in estimations related to herd size? Has this been reported in the literature to date?

Line 61-67: these are very vague statements.  How does the range of herd sizes and management styles affect what you are investigating in this study? What control strategies are you referencing that you are interested in ranking?

Materials and Methods

It was difficult to completely understand the timeline of the study – an 18-month period was mentioned but it looked like sampling on an individual farm took place over a 6-month period.  Were all farms sampled at the same time of year, especially since seasonality was identified as a limitation in the discussion? More clarity in general regarding sampling methods would be helpful.

Line 106-109: please be more explicit in exactly when young stock and adult cows were sampled, as well as when adult cows had blood samples taken and when milk samples were collected.  While this information is included in the manuscript, it took me multiple readings to understand the sampling methodology.

Line 125-151: unless this journal requires this level of detail regarding a commercial assay, this explanation is unnecessary.  Please condense and include only pertinent information that is different than what is normally recommended by the manufacturer.

Results

Many specific comments included below; overall, though, I found it hard to get a “big picture” overview of the data.  How many cows/young stock on each farm? How many new infections in each age group on each farm? Additionally, did any cows exit the study during the study period, either through culling or through drying-off?  Did any of the young stock calve and so then move into the adult cow group?

Line 199-203: it is unclear how many blood samples were from young stock and how many from adult cows – on initial reading it appears as though all samples noted in line 199 are from adult cows.  Additionally, please report the total number of young stock and adult cows tested, and whether any animals did not complete the entire study period.

Table 1: please provide data on how many animals were present on each farm, both adult cows and young stock.  It is difficult to see how the incidence rate relates to each individual farm if the total number of animals on the farm is not reported.

Figure 2: in the text you mention that the group with the highest number of new cases was the 3-4 year old cows, but in the figure this group is included with the 2-3 year old cows; recommend changing the figure to show histogram bars in 1-year increments.

Line 234-235: please include information on the number of farms performing each of these management practices.  Also, you note that these are the most frequent management practices reported; are there other practices also reported that could be associated with BLV transmission?

Line 237-239: this sentence does not make sense.

Table 2: I don’t understand this table.  The data on the left part – the age groups of cows and the number of new cases in each age group – is fairly logical, but would possibly be improved by including information on the pattern of new infections on each farm to see if farm size had an effect on either incidence rate or age of new infections.  I don’t understand how the right-hand information is related at all – what do the numbers in each column mean? For example, under AI, does the 2 in the first row mean that 2 cows in the 1-3 year age group had AI performed during the study period, or that 2 farms perform AI in cows 1-3 years old?  This data either needs to be clarified or completely removed.

Line 243-247: you have only provided one p-value for one management practice investigated; please include data on the other practices as well, if they were statistically analyzed for association with new infections.

Discussion

Most of the discussion is adequate, although I found it hard to relate all of the data presented in the results section to specific parts of the discussion.  A more detailed discussion of the difference in incidence rates between adult cows and young stock would be helpful as well.

Line 249-251: neither of your study aims in the introduction mention the objective of measuring the feasibility of monitoring BLV status through milk samples; either add this as a study objective or reword the discussion.

Line 255-257: this is the first mention of herd size for the specific herds enrolled in this study.  Please include information either in the materials and methods or the results section regarding herd size for all participating herds.

Line 271-272: did the age of new infections vary based on herd prevalence or herd size?  What is the usual lifespan of a dairy cow in Chile, and does it differ from the usual productive lifespan of a dairy cow in other countries for the studies cited in the discussion?

Line 301: I suspect the authors mean if the donor animal is persistently lymphocytotic or with a high proviral load

Line 312: another limitation that should be noted is that not all animals in the study were treated the same: the young stock were only sampled twice with blood samples, and the adult cows had multiple milk samples as well as 2 blood samples.

Round 2

Reviewer 1 Report

Im glad to see that the authors have revised the manuscript according to the reviewers comments. Some minor issues need to be clarified further.

1.line 87, it should be Serum samples were obtained from 2450 bovines,...

2.line 176, The samples number is not correct. Serum samples were taken twice (Day 0 and Day 180) from 2450 bovines, and milk was obtained from 1,527 lactating cows five times (Day 30, 60, 90, 120 and 150). Please check it.

3. line 178,180, change new cases to new infections.

4. line 180, Change positive to infected.

5. I still suggest to include the positive numbers in Table 1. I known that the main objective of Table 1 is to show Incidence rate. But the incidence rate could be related with prevalence rate in a herd. So the presentation of positive number or rate will be useful and helpful for further analysis.

This is also true for Fig.2, it seems the frequency decreased sharply for the cows >6 years old. Is it possible that the positive rate in these herds is very high so that the incidence rate of new infections was lower?

6. Line 201-202, how did the author concluded with a higher concentration between 3 and 5 years from Fig.2? It seems to be 2-6 years.

Author Response

we attached a document with our response

Reviewer 2 Report

Thank you for your revisions to the manuscript - I found it easier to follow the overall experimental protocol and understand the presentation of the results.

For Table 1, I would recommend displaying the number of new cases in the context of the number of BLV-seronegative cows, rather than the total number of cows - as the BLV-seropositive cows should not be included in the incidence calculations at all.  Keeping some indication of herd size in the table (e.g. the herd size category mentioned in the materials and methods section) will be helpful for readers to put the number of new infections in the context of the overall herd size.

I still have trouble with understanding Table 2.  For example, am I to understand that across all 11 farms, there were 20 new infections in cows aged 1-3 years, but a direct correlation with management practices could only be determined for 4 of these cases (2 AI, 2 injections)?

As only about 35% of new infections were temporally linked with the investigated management practices, it would be good to see more exploration of this and other potential confounders, bias, or other reasons why more cases could not be linked to management practices in the discussion.  The authors have spent time discussing the animals that did have a positive temporal association between management practices and new BLV infection, but not much time discussing why it was not possible to have links between management practices and infections for the other animals.

Author Response

see document attached
